# Predictors of Noninvasive Respiratory Support Failure in COVID-19 Patients: A Prospective Observational Study

**DOI:** 10.3390/medicina58060769

**Published:** 2022-06-06

**Authors:** Rolandas Zablockis, Goda Šlekytė, Rūta Mereškevičienė, Karolina Kėvelaitienė, Birutė Zablockienė, Edvardas Danila

**Affiliations:** 1Clinic of Chest Diseases, Immunology and Allergology, Institute of Clinical Medicine, Vilnius University, M.K. Ciurlionio 21, 03101 Vilnius, Lithuania; goda.slekyte@santa.lt (G.Š.); karolina.lauzikaite@santa.lt (K.K.); edvardas.danila@santa.lt (E.D.); 2Centre of Pulmonology and Allergology, Vilnius University Hospital Santaros Klinikos, Santariskiu St. 2, 08661 Vilnius, Lithuania; ruta.mereskeviciene@santa.lt; 3Clinic of Infectious Diseases and Dermatovenerology, Institute of Clinical Medicine, Vilnius University, M.K. Ciurlionio 21, 03101 Vilnius, Lithuania; birute.zablockiene@santa.lt; 4Centre of Infectious Diseases, Vilnius University Hospital Santaros Klinikos, Santariskiu St. 2, 08661 Vilnius, Lithuania

**Keywords:** COVID-19, acute hypoxemic respiratory failure, high-flow nasal cannula, non-invasive ventilation, predictive factor

## Abstract

*Background and Objective:* Respiratory assistance tactic that is best for COVID-19-associated acute hypoxemic respiratory failure (AHRF) individuals has yet to be determined. Patients with AHRF may benefit from the use of a high-flow nasal cannula (HFNC) and non-invasive ventilation (NIV). The goals of this prospective observational research were to estimate predictive factors for HFNC and NIV failure in COVID-19-related AHRF subjects. *Materials and Methods:* The research enlisted the participation of 124 patients. A stepwise treatment approach was used. HFNC and NIV were used on 124 (100%) and 64 (51.6%) patients, respectively. Thirty (24.2%) of 124 patients were intubated and received invasive mechanical ventilation. *Results:* 85 (68.5%) patients were managed successfully. Patients who required NIV exhibited a higher prevalence of treatment failure (70.3% vs. 51.6%, *p* = 0.019) and had higher mortality (59.4% vs. 31.5%, *p* = 0.001) than patients who received HFNC. Using logistic regression, the respiratory rate oxygenation (ROX) index at 24 h (odds ratio (OR) = 0.74, *p* = 0.018) and the Charlson Comorbidity Index (CCI) (OR = 1.60, *p* = 0.003) were found to be predictors of HFNC efficacy. It was the ROX index at 24 h and the CCI optimum cut-off values for HFNC outcome that were 6.1 (area under the curve (AUC) = 0.73) and 2.5 (AUC = 0.68), respectively. Serum ferritin level (OR = 0.23, *p* = 0.041) and lymphocyte count (OR = 1.03, *p* = 0.01) were confirmed as predictors of NIV failure. Serum ferritin level at a cut-off value of 456.2 ng/mL (AUC = 0.67) and lymphocyte count lower than 0.70 per mm^3^, (AUC = 0.70) were associated with NIV failure with 70.5% sensitivity, 68.7% specificity and sensitivity of 84.1%, specificity of 56.2%, respectively. *Conclusion:* The ROX index at 24 h, CCI, as well as serum ferritin level, and lymphocyte count can be used as markers for HFNC and NIV failure, respectively, in SARS-CoV-2-induced AHRF patients.

## 1. Introduction

The new coronavirus (SARS-CoV-2) became a worldwide pandemic, affecting a huge portion of the global population [1]. Numerous SARS-CoV-2 variants have been found by whole-genome sequencing of the virus. The omicron variant was linked with less severe disease in people admitted to hospital with COVID-19, whereas the alpha, beta, and delta variants were associated with more severe disease, resulting in an increased risk of admission to a high-dependency or intensive care unit, as well as significant mortality [2].

Around one-fifth of individuals with coronavirus disease (COVID-19) are classified as seriously or critically ill, having developed respiratory failure and requiring some breathing assistance [3]. When conventional oxygen treatment is ineffective, a high-flow nasal cannula (HFNC) or non-invasive ventilation (NIV) may be employed [4,5]. Using non-invasive respiratory support (NIRS) to treat patients without intubation may assist reduce healthcare resources while also lowering the risk of complications [6]. Recent research indicates that NIRS may reduce intubation rates but has no effect on mortality [7]. In contrast, delaying intubation of individuals with severe respiratory insufficiency may have negative consequences [8]. The best supportive treatment for AHRF caused by COVID-19 is still unknown. In general, the current guidelines are of poor quality and widely variable [9]. There are no clear recommendations about the indications for NIRS or which individuals will need endotracheal intubation and invasive care [10]. It is critical to identify persons who are at a greater risk of failure to avoid delays in choosing the best treatment approach [11]. The goals of this research were to identify risk variables for HFNC and NIV failure in COVID-19-related AHRF patients.

## 2. Materials and Methods

### 2.1. Study Population

From November 2020 to May 2021, we conducted prospective observational research on COVID-19-related AHRF patients admitted to our institution. The Vilnius Regional Ethics Committee (Reference Number: 158200-13-652-210, 7 July 2013) approved the research, which was determined in accordance with the Helsinki Declaration. All subjects gave their informed consent for inclusion before participated in the research.

We enrolled all consecutive COVID-19 patients in whom HFNC therapy was used. The use of NIV or invasive mechanical ventilation (IMV) prior to HFNC therapy was excluded from the study. A stepwise treatment approach was used. Respiratory support options were based on the German Respiratory Society recommended algorithm [12]. HFNC was indicated after oxygen therapy (15 L/min oxygen flow) failed. Patients received NIV when treatment with HFNC failed. Endobronchial intubation and IMV were performed when evidence of deteriorating respiratory failure on NIRS, hemodynamic instability, and tiredness was present. According to the national treatment protocol, all patients received standard treatment (antiviral medications, corticosteroids, anticoagulants, and antibiotics if needed). The concentration of C-reactive protein (CRP) in the serum was determined using the immunoturbidimetric method. For quantitating lactate dehydrogenase (LDH) activity, a photometric method according to the International Federation of Clinical Chemistry and Laboratory Medicine recommendations (ARCHITECT *c*4000 clinical chemistry analyzer by Abbott, IL, USA) was used. Interleukin 6 (IL-6) and ferritin levels were measured using the MAGLUMI 2000 Chemiluminescence Immunoassay (CLIA) System (Snibe Diagnostic, Shenzhen, China). The D-dimer test was performed using the immunoturbidimetric method (STA Compact^®^ analyzer, Diagnostica Stago, Asnières sur Seine, France).

### 2.2. Data Collection

Demographic, clinical, and therapeutic information about the research participants were obtained from their medical records. The study’s primary outcomes were the NIRS failure rate and in-hospital mortality. Patients who required intubation and IMV or died during HFNC therapy were defined as HFNC failure. The necessity for IMV or death during the hospital stay was classified as a failure of NIV.

### 2.3. Statistical Analysis

To assess differences in characteristics among the treatment groups, the Kruskal-Wallis H test was applied. For categorical data, the chi-squared test or Fisher’s exact test were used. We used logistic regression analysis to see if the independent factors had any influence on the prognosis of treatment failure. The multivariate analysis included all factors identified as predictors in the univariate analysis with a *p*-value of 0.2 or below [13]. The predictive model’s basic performance measures were calculated with the receiver operating curves (ROC) statistics analysis. A *p* < 0.05 was used as the cutoff that indicates statistical significance. SPSS version 26.0 was used to analyze the data (IBM Corp, Armonk, NY, USA).

## 3. Results

### 3.1. The Study Population’s Basic Characteristics and Outcomes

The study evaluated 127 patients and enrolled 124. Figure 1 illustrates a study flowchart.

Sixty-nine (55.6%) of the patients were male, whereas 55 (44.4%) were female. The patients’ average age was 64 years old (from 26 to 93 years). The majority of the patients (75.8%) had what was considered significant comorbidity (a CCI score of 2 or higher). The predominant chest radiological findings were bilateral (98.5%), multifocal ground-glass opacities (63.8%), and areas of consolidation (36.2%). The median PaO_2_/FiO_2_ ratio at admission was 107.0 (IQR: 86.0–133.5). A PaO_2_/FiO_2_ ratio of less than 100 was found in nearly half (41.9%) of individuals, suggesting severe hypoxemia. For a typical duration of 10 days, 96.8% of patients received nadroparin calcium subcutaneously twice daily (86 anti-Xa IU/kg body weight). Dexamethasone (at a dose of 6 mg once daily) was given to 65.3 percent of patients for up to 10 days. Remdesivir was administered intravenously to 28.2% of patients (200 mg dose on day 1, followed by a 100 mg maintenance dose administered daily on days 2 through 10). Antibiotics were given to 90.3% of the patients. HFNC therapy and NIV were administered to a total of 124 (100.0%) and 64 (51.6%) patients, respectively. Thirty-four (24.2%) of 124 patients required intubation and IMV. In total, 85 (68.5%) patients were managed successfully. Table 1 shows the demographics, clinical, laboratory data, and outcomes of patients according to different therapy groups.

Patients who required NIV were older (median: 67.0 vs. 64.0 years, *p* = 0.039), had a greater number of comorbidities, leading in a much higher CCI score (4.0 vs. 2.0, *p* = 0.012), and had a lower ROX index at 24 h (median: 5.6 vs. 6.2, *p* = 0.043) compared to subjects who underwent HFNC. Additionally, these patients had a significantly greater rate of treatment failure (70.3% vs. 51.6%, *p* = 0.019) and a considerably higher rates of death (59.4% vs. 31.5%, *p* = 0.001). In a cohort analysis of individuals treated with NIV, 22 (34.4%) of them were classified as subjects who were given a do-not-intubate (DNI) order. The reasons for DNI status were multimorbidity, severe frailty, and age. The decision was made by a multidisciplinary team, including at least an intensivist and a pulmonologist. Patients with a DNI order were older in comparison with patients without a DNI order (median age: 75.5 vs. 63.5 years, *p* < 0.001), had more comorbidities (CCI score: 6.0 vs. 2.0 points, *p* < 0.001), had lower PaO_2_/FiO_2_ (81.5 vs. 113.5, *p* = 0.007), lower ROX index at 24 h (median: 4.5 vs. 6.0, *p* = 0.008) and a greater rate of mortality (72.7% vs. 52.4%, *p* = 0.035). When individuals who had a DNI order were matched to individuals who did not have a DNI order, the time of hospitalization was considerably shorter (median: 15.0 vs. 24.0 days, *p* = 0.019).

### 3.2. Predictors of HFNC and NIV Failure

The binary logistic regression analysis of risk variables for HFNC and NIV failure is shown in Table 2 and Table 3.

The regression model for HFNC failure was significant (χ^2^ = 15.9, *p* = 0.043) and accurately predicted 82.0% of cases. The Nagelkerke’s r-squared statistic value was 0.45. In the univariate analysis that was conducted, age, comorbidities, and the ROX index were all shown to be significant predictors of HFNC failure. In the multivariate analysis, a number of covariates such as age, comorbidities, the PaO_2_/FiO_2_ ratio, and the ROX index were included. The multivariate analysis confirmed that the only ROX index at 24 h (OR = 0.74; 95% CI = 0.58–0.95, *p* = 0.018) and CCI (OR = 1.60; 95% CI = 1.18–2.18, *p* = 0.003) were predictors of HFNC efficacy.

It was shown that the regression model for NIV failure was statistically significant (χ^2^ = 52.7, *p* < 0.001) and that it predicted 84.6% of events accurately. The Nagelkerke’s r-squared statistic value was 0.67. The univariate analysis showed that only obesity was a significant predictor of NIV failure. The mulivariate analysis comprised six variables: gender, obesity, ROX index, lymphocyte count, ferritin levels, and Remdesivir usage. Serum ferritin level (OR = 0.23; 95% CI = 0.10–0.86, *p* = 0.041) and lymphocyte count (OR = 1.03; 95% CI = 1.01–1.05, *p* = 0.015) were confirmed as predictors of NIV failure in the multivariate logistic regression model.

### 3.3. The Predictive Model’s Performance

Table 4, Figure 2 and Figure 3 show the ROC curves statistical data and the ROC curves of the prediction for HFNC and NIV failure.

The AUC of the CCI and ROX index at 24 h for predicting HFNC failure was 0.73 (*p* < 0.001) and 0.68 (*p* < 0.001), respectively. The AUC difference between the CCI and ROX index at 24 h under the ROC curves was nonsignificant (Z test value = 0.641, AUC difference 0.042, *p* = 0.522).

With a cut-off value of 2.5, the best cut-off point for CCI results in a sensitivity of 64.1% and a slightly higher specificity of 75.0%. The optimum cut-off value of the ROX index at 24 h for HFNC outcome was 6.1 (sensitivity of 81.2% and specificity of 51.7%*).*

Serum ferritin (cut-off value: 456.2 ng/mL) had a moderate sensitivity (70.5%), a specificity of 68.7%, and an AUC of 0.67 (*p* = 0.037) in relation to the NIV outcome. NIV failure, according to the data, was related with a lymphocyte count of less than 0.70 per mm^3^ with an 84.1% sensitivity, a 56.2% specificity, and an AUC of 0.70 (*p* = 0.009). The AUC difference between serum ferritin level and lymphocyte count under the ROC curves was nonsignificant (Z test value = −0.230, AUC difference −0.027, *p* = 0.818).

## 4. Discussion

The goal of this research was to determine predictors for NIRS failure in COVID-19-related AHRF subjects. The study’s major findings were as follows: (1) The failure rates of HFNC and NIV as rescue therapy were 51.6% and 70.3%, respectively (2) 24.2% of patients required intubation and were treated with IMV (3) In-hospital mortality rate was 31.5% overall (4) The ROX index at 24 h and the CCI were both predictive of HFNC effectiveness, and (5) ferritin and lymphocyte count levels were found to be significant predictors of NIV failure.

In individuals with AHRF, oxygen therapy is the initial treatment option [14]. However, if hypoxemia persists with increased dyspnea and respiratory rate, NIRS should be prescribed [6]. If there are no immediate reasons to intubate, HFNC might be used to improve oxygenation [14].

NIV may be utilized as a first-line treatment as well as a follow-up therapy after HFNC [9]. NIV showed variable success rates, from 11% in milder cases to 80% in severely ill patients [6,9,15]. The HFNC therapy and NIV failure rates in this study were in concordance with previously reported data [3,7,11].

The impact of time on intubation for severely ill patients with AHRF remains a topic of debate. Some physicians advocate early intubation, while others have used a stepwise strategy that includes NIV or HFNC before intubation [16]. Recent studies showed that intubation time in COVID-19-related AHRF patients is unrelated to death [6,17,18]. If the initial NIV patient fails, the chance of death is the same as for those who are intubated first (84% vs. 82%) [19].

The overall mortality rate observed in our study does not differ from those found in several previous studies [20,21]. In-hospital mortality was reduced when we removed individuals having DNI orders (with DNI orders: 72.7% vs. without DNI orders: 22.5%). In our research, patients with a DNI order were substantially older than patients without a DNI order, had more comorbidities, had lower PaO_2_/FiO_2_ ad entry, and had a poorer ROX index at 24 h. We hypothesized that the greater mortality in DNI patients was related to ineffective NIV therapy for these patients, who had more comorbidities and more severe COVID-19 illness.

The ROX index is frequently utilized as a promising instrument for detecting failure of HFNC treatment in COVID-19 pneumonia individuals [12,22]. As per a meta-analysis, the ROX index within 24 h after admission with a cut-off value of 5.0 might predict HFNC failure with a sensitivity of 70.0% and a specificity of 79.0% [23]. In accordance with this, our data show that the ROX index is a good way to predict the failure of HFNC treatment 24 h after it begins, with a cut-point value of 6.1, giving the ROX index more sensitivity (81.2%) but less specificity (51.7%).

Comorbidities may have an impact on outcomes in individuals who have more severe illnesses. The Charlson Comorbidity Index (CCI) is the most often used index to assess long-term prognosis and mortality [24]. A higher CCI is linked to illness severity and an increased likelihood of death [25]. However, not all comorbidities are associated with the same level of risk [26]. Obesity is the most common comorbidity, and respiratory illness is the most strongly predictive of a severe or fatal outcome. Having a better understanding of these risk variables might assist doctors in better identifying and managing high-risk groups [27,28,29,30]. We suggest that a higher CCI predicts HFNC failure, which might be due to the fact that patients with numerous comorbidities seem to be at greater risk of developing severe disease, leading to unfavorable outcomes [31].

We discovered that the CCI and the ROX index at 24 h had a modest predictive ability for predicting HFNC treatment failures (AUC: 0.73 vs. AUC: 0.68). Additionally, they observed that the CCI index had greater specificity than the ROX index at 24 h (75.0% vs. 51.7%). We propose that the CCI index be employed in addition to the ROX index at 24 h to forecast the HFNC’s failure, which would provide more safety for the patient, though this would imply a closer clinical observation of the person.

The fact that serum ferritin and lymphocyte counts were shown to be predictive of NIV failure was an interesting discovery in this research. These results imply that ferritin levels and lymphopenia may be used as biomarkers to determine individuals most at risk of having their NIV fail and may contribute to the early development of therapeutic alternatives.

Elevated ferritin levels have indeed been to a variety of inflammatory and infectious illnesses. Increased ferritin levels were linked to higher mortality and the requirement for mechanical ventilation [32,33]. In their study, Carubbi et al., demonstrated that greater ferritin levels are related to more severe lung damage as seen by computed tomography [34]. Serum ferritin has been identified as a biomarker of both acute and chronic inflammation. A substantial increase in serum ferritin may signal activation of the monocyte-macrophage system, which is [35] an important component of the inflammatory cytokine storm. Individuals with COVID-19 who are in a severe stage often die as a consequence of a cytokine storm, which represents an abnormally uncontrolled inflammatory response [35]. In fact, hyperferritinemia is a necessary condition for diagnosing macrophage activation syndrome (MAS) [36,37]. MAS may cause broad tissue damage and multi-organ failure [38]. Hyperferritinemia is a possible marker for distinguishing a subgroup of hospitalized patients with viral infection who are at increased risk of getting poor outcomes such as respiratory failure or in-hospital death [39,40]. A critical issue is whether ferritin is a mediator or a consequence of inflammation. Several prior studies, however, have revealed ferritin’s pro-inflammatory and immunosuppressive effects, which may vary depending on the setting and the numerous signaling pathways triggered [35,41,42]. 

Lymphopenia is indeed a frequent occurrence in COVID-19 individuals and may play an essential part in disease severity [43]. Lymphopenia causes delayed virus clearance, a shift in adaptive immune responses to inflammatory responses, subsequent activation of macrophages and neutrophils, and excessive cytokine synthesis, both of which induce multi-organ damage and may prove fatal. There are a variety of processes that might result in the onset of lymphopenia by altering lymphocyte production as well as shortening survival. Lymphocyte accumulation in the lungs and other organs, including the gastrointestinal system, liver, and kidneys, has been proposed as explanation for the decrease in blood lymphocytes [44,45,46]. Significant lymphopenia increases the chance of developing adverse events [47]. People who have lymphopenia have a higher possibility of being admitted to the critical care unit and dying, especially if their lymphocyte counts stay low for more than a week after they start having symptoms [48]. Lymphocyte count analysis may help doctors find people who are most likely to get sick and start treatment early to improve their chances of getting better [49].

One of the advantages of the research is that it was conducted in a prospective way. The study population mostly included patients with moderate or severe hypoxemia at admission. However, our research has certain limitations. Firstly, this research was done at a single hospital. Second, the study’s findings may be influenced by the study’s small sample size. Third, ventilator parameters could not be analyzed, and this may have a prognostic role in the outcomes.

## 5. Conclusions

The ROX index at 24 h, CCI, as well as serum ferritin level, and lymphocyte count can be used as markers for HFNC and NIV failure, respectively, in SARS-CoV-2-induced AHRF patients.

## Figures and Tables

**Figure 1 medicina-58-00769-f001:**
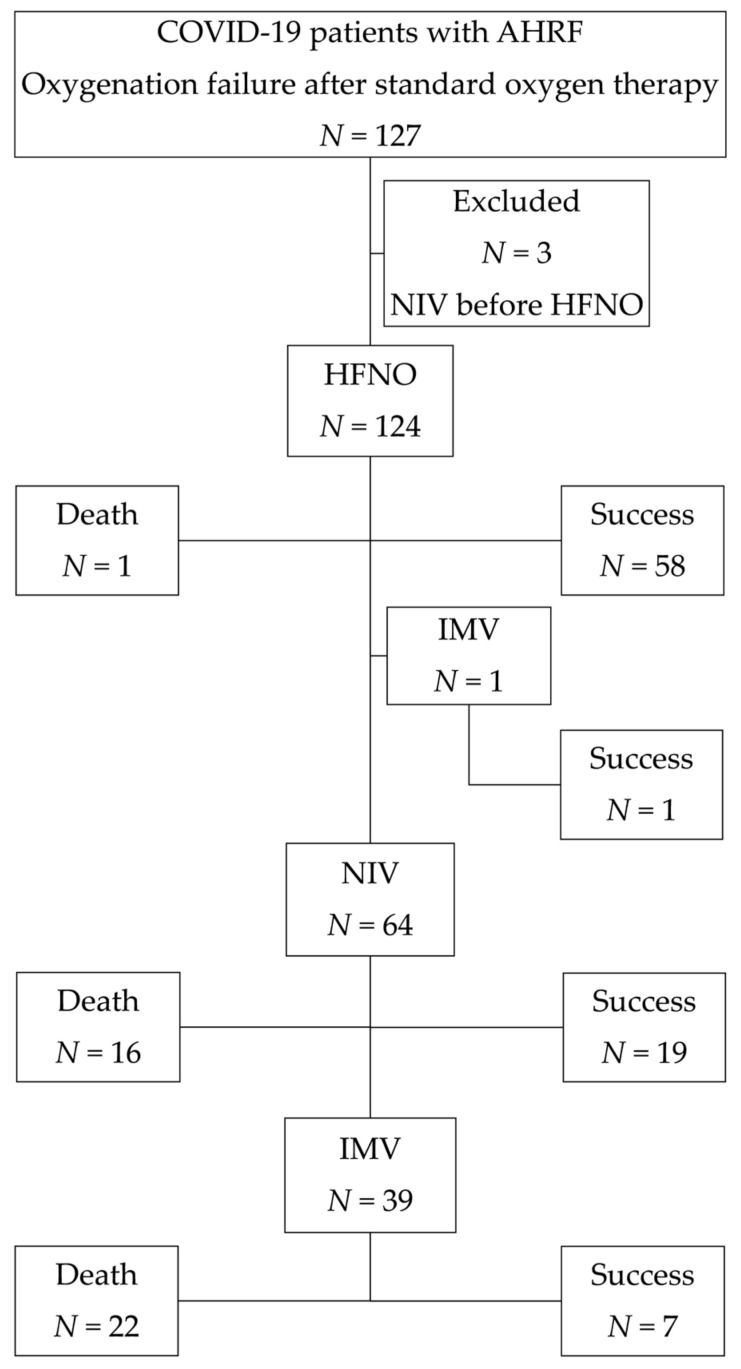
The flow of screening and patient inclusion. AHRF: acute hypoxemic respiratory failure; HFNC: high-flow nasal cannula; NIV: non-invasive ventilation; IMV: invasive mechanical ventilation.

**Figure 2 medicina-58-00769-f002:**
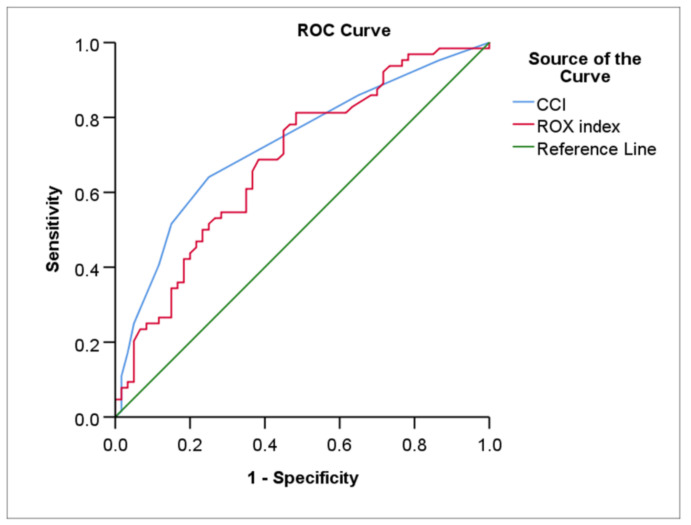
The ROC curves of the CCI and ROX index at 24 h for predicting HFNC failure. HFNC: high-flow nasal cannula; CCI: Charlson comorbidity index; ROX: the respiratory rate oxygenation; ROC: Receiver operator characteristic.

**Figure 3 medicina-58-00769-f003:**
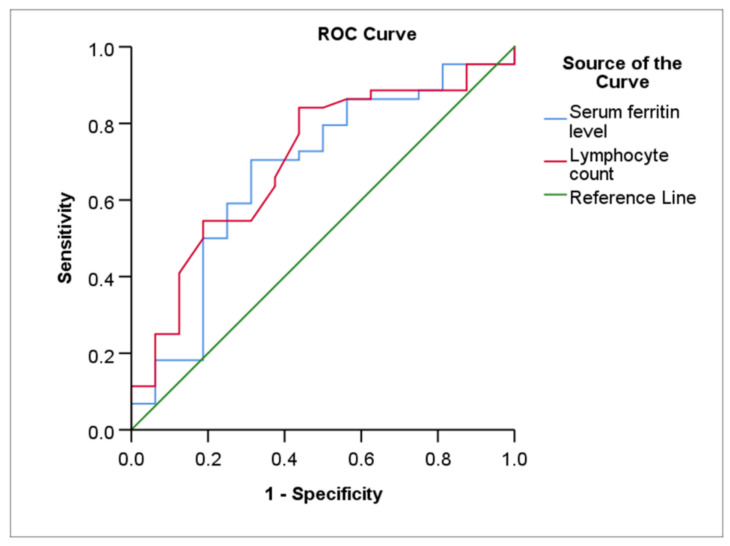
The ROC curves of serum ferritin level and lymphocyte count for prediction of NIV failure. NIV: non-invasive ventilation; ROC: Receiver operator characteristic.

**Table 1 medicina-58-00769-t001:** Patients’ characteristics and results differed depending on their therapy group.

Characteristics	HFNC (*N* = 124)	NIV (*N* = 64)	*p*-Value
Age, years	64.0 (57.0–70.0)	67.0 (61.2–74.0)	**0.039**
Gender, male, %	69 (55.6)	34 (53.1)	0.759
BMI > 30, kg/m^2^, %	17 (13.7)	10 (15.6)	0.827
CCI	2.0 (2.0–5.0)	4.0 (2.0–5.7)	**0.012**
PaO_2_/FiO_2_	107.0 (86.0–133.5)	102.5 (78.2–127.0)	0.344
ROX index at 24 h	6.2 (4.9–7.9)	5.6 (4.4–6.8)	**0.043**
WBC, count per mm^3^	6.7 (5.3–9.1)	6.8 (5.5–9.6)	0.910
Lymphocyte, count per mm^3^	0.8 (0.5–1.1)	0.8 (0.5–1.0)	0.679
NLR	6.6 (4.0–10.4)	6.8 (4.4–11.9)	0.660
CRP, mg/dL	124.2 (63.4–182.2)	131.2 (94.5–187.4)	0.252
Ferritin, ng/mL	902.0 (378.0–2118.4)	581.9 (316.9–1927.6)	0.317
IL-6, pg/mL	47.8 (19.4–96.7)	51.7 (30.7–106.7)	0.301
LDH, IU/L	462.0 (319.0–594.0)	481.0 (334.7–620.2)	0.702
D-dimer, ng/mL	660.0 (490.0–995.0)	717.5 (527.5–1210.0)	0.258
Dexamethasone, %	81 (65.3)	42 (65.6)	0.613
Remdesivir, %	35 (28.2)	18 (28.1)	0.583
Antibiotics, %	112 (90.3)	59 (92.2)	0.792
LMWH, %	120 (96.8)	62 (96.9)	0.668
The number of days since the start of symptoms	7.0 (5.0–9.0)	7.0 (5.0–9.0)	0.492
Treatment failure, %	64 (51.6)	45 (70.3)	**0.019**
In-hospital mortality, %	39 (31.5)	38 (59.4)	**0.001**
Hospitalization duration, days	21.0 (13.2–30.0)	20.5 (13.2–31.0)	0.875

Data are shown as the median with an interquartile range or as number (%). Significant values are shown in bold. HFNC: high-flow nasal cannula; NIV: non-invasive ventilation; ROX: the respiratory rate oxygenation; BMI: body mass index; CCI: Charlson comorbidity index; PaO_2_/FiO_2_: arterial-to-inspired oxygen ratio; WBC: white blood cell; NLR: neutrophil-to-lymphocyte ratio; CRP: C reactive protein; IL-6: interleukin 6; LDH: lactate dehydrogenase; LMWH: low-molecular-weight heparin.

**Table 2 medicina-58-00769-t002:** HFNC failure predictors.

Characteristics	Univariate Analysis		Multivariate Analysis	
	OR (95% CI)	*p*-Value	OR (95% CI)	*p*-Value
Age, years	1.05 (1.02–1.09)	**0.002**	0.98 (0.93–1.03)	0.431
Gender, male	0.81 (0.39–1.65)	0.560		
BMI > 30, kg/m^2^	1.40 (0.49–3.96)	0.523		
CCI	1.52 (1.24–1.86)	**0.001**	1.60 (1.18–2.18)	**0.003**
PaO_2_/FiO_2_	0.99 (0.95–1.00)	0.078	1.00 (0.99–1.01)	0.405
ROX index at 24 h	0.77 (0.65–0.92)	**0.003**	0.74 (0.58–0.95)	**0.018**
WBC count, per mm^3^	1.02 (0.93–1.13)	0.632		
Lymphocyte count, per mm^3^	0.88 (0.48–1.62)	0.692		
NLR	1.03 (0.96–1.09)	0.405		
CRP, mg/dL	1.00 (1.00–1.01)	0.074	1.00 (0.99–1.01)	0.198
Ferritin, ng/mL	1.00 (1.00–1.01)	0.700		
IL-6, pg/mL	1.00 (0.98–1.01)	0.292		
LDH, IU/L	1.00 (0.99–1.01)	0.438		
D-dimer, ng/mL	1.00 (0.99–1.01)	0.203		
Dexamethasone	1.08 (0.25–4.60)	0.920		
Remdesivir	1.07 (0.99–1.08)	0.719		
Antibiotics	0.64 (0.19–2.14)	0.471		
LMWH	1.07 (0.15–7.84)	0.948		
The number of days since the start of symptoms	0.97 (0.88–1.07)	0.502		

The data are reported as an odds ratio (OR) with a confidence interval (CI). Significant values are shown in bold. HFNC: high-flow nasal cannula; BMI: body mass index; CCI: Charlson comorbidity index; PaO_2_/FiO_2_: arterial-to-inspired oxygen ratio; ROX: the respiratory rate oxygenation; WBC: white blood cell; NLR: neutrophil-to-lymphocyte ratio; CRP: C-reactive protein; IL-6: interleukin 6; LDH: lactate dehydrogenase; LMWH: low-molecular-weight heparin.

**Table 3 medicina-58-00769-t003:** NIV failure predictors.

Characteristics	Univariate Analysis		Multivariate Analysis	
	OR (95% CI)	*p*-Value	OR (95% CI)	*p*-Value
Age, years	1.00 (0.96–1.05)	0.733		
Gender, male	2.57 (0.85–7.77)	0.094	1.73 (0.46–6.46)	0.262
BMI > 30, kg/m^2^	4.73 (1.15–19.4)	**0.031**	0.10 (0.10–2.32)	0.096
CCI	1.11 (0.87–1.41)	0.384		
PaO_2_/FiO_2_	0.99 (0.97–1.01)	0.243		
ROX index at 24 h	0.83 (0.65–1.06)	0.145	0.63 (0.25–1.64)	0.540
WBC count, per mm^3^	0.93 (0.81–1.06)	0.256		
Lymphocyte count, per mm^3^	0.55 (0.23–1.29)	0.172	0.23 (0.10–0.86)	**0.041**
NLR	1.04 (0.95–1.14)	0.366		
CRP, mg/dL	0.99 (0.99–1.01)	0.268		
Ferritin, ng/mL	1.00 (0.99–1.01)	0.184	1.03 (1.01–1.05)	**0.015**
IL-6, pg/mL	0.99 (0.99–1.01)	0.506		
LDH, IU/L	1.00 (0.99–1.01)	0.754		
D-dimer, ng/mL	1.00 (0.99–1.01)	0.417		
Dexamethasone	0.78 (0.07–7.99)	0.833		
Remdesivir	0.39 (0.12–1.24)	0.111	0.43 (0.03–6.92)	0.554
Antibiotics	0.61 (0.09–3.96)	0.602		
LMWH	1.00 (0.99–1.01)	0.990		
The number of days since the start of symptoms	0.93 (0.81–1.06)	0.250		

Data are reported as odds ratio (OR) with confidence interval (CI). Significant values are shown in bold. NIV: non-invasive ventilation; BMI: body mass index; CCI: Charlson comorbidity index; PaO_2_/FiO_2_: arterial-to-inspired oxygen ratio; ROX: the respiratory rate oxygenation; WBC: white blood cell; NLR: neutrophil-to-lymphocyte ratio; CRP: C reactive protein; IL-6: interleukin 6; LDH: lactate dehydrogenase; LMWH: low-molecular-weight heparin.

**Table 4 medicina-58-00769-t004:** ROC curve data of the predictive model for HFNC and NIV failure.

Treatment Group	Characteristics	Sensitivity (%)	Specificity (%)	Cut-Off Value	AUC (95% CI)	*p*-Value
HFNC (*N* = 124)	CCI	64.1	75.0	2.5	0.73 (0.64–0.82)	**<0.001**
	ROX index at 24 h	81.2	51.7	7.1	0.68 (0.59–0.78)	**<0.001**
NIV (*N* = 64)	Lymphocyte count, per mm^3^	84.1	56.2	1.0	0.70 (0.55–0.85)	**0.009**
	Ferritin, ng/mL	70.5	68.7	456.2	0.67 (0.51–0.84)	**0.037**

Significant values are shown in bold. HFNC: high-flow nasal cannula; NIV: non-invasive ventilation; CCI: Charlson comorbidity index; ROX: the respiratory rate oxygenation; AUC: area under the ROC curve; CI: confidence interval.

## Data Availability

The data presented in this study are available on request from the corresponding author. The data are not publicly available due to data protection policies.

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
