# Peer review of "Predictors of Noninvasive Respiratory Support Failure in COVID-19 Patients: A Prospective Observational Study"

_medicina, 2022, doi:10.3390/medicina58060769_

Round 1

Reviewer 1 Report

In this prospective study, the authors aimed to assess predictive factors for high-flow nasal cannula (HFNC) and non-invasive ventilation (NIV) failure in 124 COVID-19-associated acute hypoxemic respiratory failure (AHRF) subjects. HFNC and NIV were used on 124 (100%) and 64 (51.6%) patients, respectively. Thirty (24.2%) of 124 patients were intubated and received invasive mechanical ventilation.

They found that 85 (68.5%) patients were managed successfully. Patients who required NIV exhibited a higher prevalence of treatment failure (70.3% vs. 51.6%, p = 0.019) and had higher mortality (59.4% vs. 31.5%, p = 0.001) than patients who received HFNC. Using logistic regression, the ROX index at 24h [OR= 0.74, p = 0.018] and the Charlson Comorbidity Index (CCI) [OR = 1.60, p=0.003)] were found to be predictors of HFNC efficacy. It was the ROX index at 24 h and the CCI optimum cut-off values for HFNC outcome that were 6.1 [area under the curve (AUC) = 0.73] and 2.5 (AUC = 0.68), respectively. Serum ferritin level (OR = 0.23, p = 0.041) and lymphocyte count (OR = 1.03, p = 0.01) were confirmed as predictors of NIV failure. Serum ferritin level at a cut-off value of 456.2 ng/ml (AUC = 0.67) and lymphocyte count lower than 0.70 per mm3 , (AUC = 0.70) were associated with NIV failure with 70.5% sensitivity, 68.7% specificity and sensitivity of 84.1%, specificity of 56.2%, respectively.

They concluded that the ROX index at 24 h, CCI, serum ferritin level, and lymphocyte count could be used in SARS-CoV-2-induced AHRF patients as markers for HFNC and NIV failure, respectively.

The study is of interest and of clinical impact. However, some points deserve further data and should be addressed.

-study population's basic characteristics and outcomes: the authors should also report radiologic features of covid pulmonary disease involvment (limited? diffuse? bilateral?)

-Discussion: the authors discussed the potential use of serum ferritin level and lymphocyte count as markers for HFNC and NIV failure. Regarding the serological marker of prognostic significance, the authors should recall the previously reported prognostic significance of positivity for antinuclear autoantibodies (particularly those exhibiting the "nucleolar" pattern), as recently demonstrated (COVID-19 and Immunological Dysregulation: Can Autoantibodies be Useful? Clin Transl Sci. 2021 Mar;14(2):502-508.).

Author Response

Thank you very much for your suggestions and remarks.

Point 1: Study population's basic characteristics and outcomes: the authors should also report radiologic features of covid pulmonary disease involvement (limited? diffuse? bilateral?)

Response 1:

Point 1. Radiological findings of COVID-19 pulmonary disease were added to the results.

Point 2: Discussion: the authors discussed the potential use of serum ferritin level and lymphocyte count as markers for HFNC and NIV failure. Regarding the serological marker of prognostic significance, the authors should recall the previously reported prognostic significance of positivity for antinuclear autoantibodies (particularly those exhibiting the "nucleolar" pattern), as recently demonstrated (COVID-19 and Immunological Dysregulation: Can Autoantibodies be Useful? Clin Transl Sci. 2021 Mar;14(2):502-508.).

Response 2:

Point 2. Pascolini, S., Vannini, A., Deleonardi, G., Ciordinik, M., Sensoli, A., Carletti, I., Veronesi, L., Ricci, C., Pronesti, A., Mazzanti, L., Grondona, A., Silvestri, T., Zanuso, S., Mazzolini, M., Lalanne, C., Quarneti, C., Fusconi, M., Giostra, F., Granito, A., Muratori, L., … Muratori, P. (2021). COVID-19 and Immunological Dysregulation: Can Autoantibodies be Useful? Clinical and translational science14(2), 502–508. https://doi.org/10.1111/cts.12908.

This article did not examine the population of patients receiving HFNO therapy or NIV respiratory support for respiratory failure due to severe COVID-19 infection. Therefore, we did not include this article in the discussion of the results of our study.

Reviewer 2 Report

Hello,

Thank you conducting the this prospective study in Covid 19 patients at the time of hospitalization. The study conducted and reported well. The manuscript is good shape with well written, so well done to authors. Please see below minor comments.

Minor

1. It would be better representation if author can change the font style of Figure 1 to be same as the manuscipt.

Author Response

Thank you very much for your suggestions and remarks.

Point 1: 1. It would be better representation if author can change the font style of Figure 1 to be same as the manuscipt.

Response 1:

Point 1. We changed the font style of Figure 1 to be the same as the manuscript.

Reviewer 3 Report

Dear Authors, Congratulations on the study and the analysis. Prospective study provides high quality data and the Charlson Comorbidity/ROX Index you used makes results possible to compare with other studies. If it is possible, please write more details about what doses of LMWH were used and what kind of test was used to measure ferritine. It would be useful for future analysis.

Author Response

Thank you very much for your suggestions and remarks.

Point 1: 1. If it is possible, please write more details about what doses of LMWH were used and what kind of test was used to measure ferritin. It would be useful for future analysis.

Response 1:

Point 1. We described the doses of LMWH administered and the tests used to determine serum levels of CRP, D-dimmers, ferritin, IL-6, and LDH.